# Sexually Explicit Online Media Use and Sexual Behavior among Sexual Minority Men in Portugal

**DOI:** 10.3390/bs11030038

**Published:** 2021-03-18

**Authors:** Henrique Pereira, Graça Esgalhado

**Affiliations:** 1Department of Psychology and Education, Faculty of Social and Human Sciences, University of Beira Interior, Pólo IV, 6200-209 Covilhã, Portugal; mgpe@ubi.pt; 2Health Sciences Research Centre (CICS-UBI), 6200-506 Covilhã, Portugal; 3Research Centre in Sports Sciences, Health Sciences and Human Development (CIDESD), 5001-801 Vila Real, Portugal; 4Institute of Cognitive Psychology, Human and Social Development (IPCDHS), 3000-115 Coimbra, Portugal

**Keywords:** sexually explicit online media (SEOM), sexual behavior, men who have sex with men (MSM), HIV, Portugal

## Abstract

*Introduction*: This study aimed to describe sexually explicit online media (SEOM) use among men who have sex with men (MSM) in Portugal and to examine any associations between exposure to SEOM depicting unprotected anal intercourse and engaging in unprotected anal sex. *Methods*: This study recruited 1577 MSM living in Portugal with Internet access, who ranged in age from 18 to 74 years old (*M_age_* = 35.69, *SD_age_* = 11.16). Participants were recruited via websites, e-mail and social media. 67.3% (n = 1061) of sample participants self-identified as gay, and 32.7% (n = 516) claimed to be bisexual. The survey included four categories of questions/measurements, encompassing demographic information, SEOM use, explicit imagery of protected/unprotected anal sex and sexual behavior. *Results*: The study results suggest that Portuguese MSM frequently use SEOM and that they possess a stated preference for SEOM displaying unprotected anal sex. Furthermore, this study’s findings indicate that self-identified gay men more frequently engage in unprotected sex than self-identified bisexual men. Finally, the study revealed that a preference for viewing SEOM displaying unprotected sex and higher levels of arousal attributed to direct SEOM exposure are significant predictors of having receptive anal sex without condoms.

## 1. Introduction

The Internet has undoubtedly facilitated access to sexually explicit online media (SEOM) [1,2]. This type of content is usually displayed through the description or exposure of sexual organs and/or sexual activities to consumers who view it [3,4]. SEOM depicting men who have sex with men (MSM) is mostly made by men, particularly by MSM [5]. Many believe that viewing SEOM is related to factors associated with the formation of norms and social roles [6]. However, most men simply state that they view SEOM when they are sexually aroused, want to reach orgasm or are bored [7]. Technology has allowed for easier and more anonymous SEOM access, without limitations on its consumption, and while offering a variety of possibilities for its use through videos, live broadcasts, pictures and erotic stories [8,9]. Numbers vary widely depending on culture, year, sampling technique, and definition of SEOM, but various international studies have put online pornography use rates at 50–99% among men [3,4]. Thus, SEOM plays an important role in shaping knowledge, attitudes and behavior [10] in the lives of MSM.

Research shows that SEOM exposure possesses a significant association with several variables, such as relationship quality [11], personality factors (anxiety, fear of rejection and low self-esteem) [12], risky sexual behaviors [1,2,4,5,9], mental health and body image [13], and aggressive and degrading behaviors [14], showing that motivations behind SEOM use can be broken down into four factors—relationship, mood management, habitual use, and fantasy. The fact that viewing SEOM online has become common for an increasing number of individuals has motivated researchers to document the different profiles of SEOM consumers, finding that some 3% to 12% of users possess compulsive profiles associated with lower levels of sexual satisfaction and performance and higher levels of sexual compulsivity and avoidance [15,16].

MSM (usually self-identified gay or bisexual men), indicate significantly more frequent use of SEOM (96% to 98%) compared to heterosexual men (80%), as well as more frequent viewing of condomless anal sex [17], which has been consistently associated with greater exposure to risky sexual practices and sexual dysfunctions [18,19,20]. As a result, it is clear that paying greater attention to the mediating role of SEOM in the study of sexual practices among MSM is an important task with relevant implications for health and social interventions. This finding is especially pertinent in Portugal, a mainly Catholic, Southern European country, where no research into this topic has been conducted to date.

Despite the fact that Portugal currently enjoys an unprecedented degree of political and social acceptance of same-sex relations, ranking among the top six European Union countries with respect to human rights and full equality for lesbian, gay, bisexual and trans (LGBT) people [21], Portuguese MSM possess one of the highest HIV prevalence rates in Western Europe [22] and continue to be one of the most at-risk groups for HIV transmission, since the number of new HIV infections in this group represented 22.8% of all HIV transmissions in the country [23,24]. Therefore, it is pertinent to explore less obvious risk factors (such as SEOM use) that are associated with higher risk sexual behaviors. Since so little is known regarding the influence of SEOM use on MSM’s sexual practices and no relevant studies currently exist in Portugal, the present study sought to answer the following research questions: What are SEOM consumption levels and motives, sexual practices and the association between SEOM consumption and sexual practices among MSM?

## 2. Materials and Methods

### 2.1. Measures 

The survey included four categories of questions/measurements, encompassing sociodemographic information, SEOM use, explicit imagery of protected/unprotected anal sex and sexual behaviors.

### 2.2. Sociodemographic Information

Items included age, gender, marital status, place of residence, educational attainment, professional status, socioeconomic status, weekly frequency of sexual intercourse and sexual orientation.

### 2.3. SEOM

Participants’ perceived levels of SEOM use were measured by asking the following questions: “How often do you use SEOM?” (responses ranged from 1–2 days a week to everyday); “On average, how much time do you spend viewing SEOM?”; “What are your motives for using SEOM?”; “What portion of SEOM that you consume specifically depicts anal sex without condoms (bareback sex)?”; and “What are your motives for viewing bareback SEOM?” Participants were asked to choose from a set of multiple-choice motives such as “because I am very sexually aroused”, “because I want to release sexual arousal”, “to entertain myself because I am bored”, “to relieve stress”, “because it is a habit or a routine”, among other motives.

### 2.4. Explicit Imagery of Protected/Unprotected Anal Sex

Two pictures portraying anal sex were displayed. Picture A showed the insertion of a penis into a man’s anus with a condom, while picture B showed the insertion of penis into a man’s anus without a condom. Participants were asked to rate their levels of sexual arousal for both pictures using a 10-point scale, ranging from 1 (not at all aroused) to 10 (completely aroused). Both pictures depicted similar positions, similar body types and similar penis shapes, with the only difference involving condom usage.

### 2.5. Sexual Behavior

Participants were asked to recall their sexual behavior during the previous month. The study collected information regarding the frequency of receptive and insertive anal sex, as well as condom use during sexual intercourse.

### 2.6. Procedures

Recruitment consisted of online notifications (emails and electronic messages) and advertisements sent to LGBT community organizations, mailing lists, and social networks, such as Facebook. Inclusion criteria were comprised of being 18 years of age or older, using SEOM, understanding Portuguese and consenting to being exposed to sexually explicit content during the survey. Participants responded to the study’s outreach online through a website created for this purpose. All advertisements referred participants directly to the online survey, where they were informed that their responses would be anonymous and confidential, in accordance with the Helsinki Declaration of ethical principles concerning research involving human subjects. In addition, this study was also approved by the researchers’ university’s ethics committee. The first page of the questionnaire explained the study’s objectives and informed participants about how to complete the survey, their freedom to withdraw from the study at any time and how to contact the author for further information about the study, if needed. Confidentiality was ensured by using codes on documents containing study data, by encrypting identifiable data, by assigning security codes to computerized records and by limiting access to identifying information (e.g., IP addresses).

## 3. Results

### 3.1. Sociodemographic Information

The study sample was composed of 1577 MSM who view SEOM, ranging in age from 18 to 74 years old (*M_age_* = 35.69, *SD_age_* = 11.16), who were recruited via websites, e-mail and social media to participate in the study. Among the entire sample, 67.3% (n = 1061) self-identified as gay, and 32.7% (n = 516) claimed to be bisexual. The sample participants were sexually active, claiming to have sex 3.21 times per week, on average. Additionally, the majority of the overall sample did not have a partner, held a university degree, and lived in urban areas. Study participants’ sociodemographic information is described in greater detail in Table 1.

### 3.2. SEOM Use

All participants reported using SEOM, ranging from 1 to 2 times per week (25%) to everyday (36%), although a plurality of participants claimed to use SEOM 3 to 6 times per week (39%). Participants’ reported SEOM viewing sessions ranged from 5 to 240 min long, lasting a mean of 70.88 min (SD = 124.70) per viewing session.

Table 2 shows the results for SEOM use according to sexual orientation. Regarding motives for SEOM use, participants most frequently mentioned reasons for viewing SEOM were the result of sexual arousal, wanting to release sexual arousal, to be entertained because of boredom, to relieve stress and because of habits/routine actions, with the latter explanation being significantly more common among self-identified bisexual men. Regarding SEOM portraying anal sex without condoms, participants reported that an average of 67% of their SEOM use involved bareback content. The most mentioned reasons for viewing bareback content were that it looked more natural, it reminded participants of what sex could be like in the absence of sexually transmitted infections and it stimulated higher levels of sexual arousal. Significant differences were found when comparing gay and bisexual men’s consumption of bareback content. Self-identified gay men were more likely than bisexual men to state that bareback SEOM was similar to their real life sexual practices and that they enjoyed watching SEOM involving anal sex, with or without condoms.

Table 3 describes SEOM use by sexual orientation. Regarding the activities that participants engaged in after viewing SEOM, 84.1% of participants mentioned masturbation. Concerning the preferred mediums of SEOM consumption, the vast majority of participants (82.9%) claimed that videos were their primary source of SEOM. Furthermore, 47.4% of all participants said that they had no preference for SEOM depicting safe or unsafe sex.

### 3.3. Anal Sexual Behavior

The most common anal sexual activities reported by study participants were receptive and insertive anal sex with condoms. Table 4 shows these results by sexual orientation, indicating that self-identified gay men were more likely to partake in unprotected sex than self-identified bisexual men.

### 3.4. Exposure to Sexually Explicit Content and Anal Sexual Behavior

Participants attributed statistically significant [t(159) = 37.730; *p* =< 0.001] lower levels of arousal to picture A (anal sex with a condom) (6.54; SD = 2.18) than to picture B (anal sex without a condom) (7.59; SD = 2.76). Table 5 displays the association between exposure to explicit content (pictures A and B) and anal sexual behavior. Positive and significant correlations were observed between both receptive and insertive anal sex without a condom and levels of arousal due to exposure to picture B, which portrayed bareback sex.

Finally, the study conducted a hierarchical multiple regression analysis predicting receptive anal sex without condoms. Table 6 displays the results concerning the effects of SEOM use on partaking in receptive anal sex without condoms. The possible confounding variables “age” and “preference for SEOM without condoms” were added to the first block. The length of SEOM viewing sessions was added to the second block. Finally, exposure to images A and B was added to the third block. The first block of the analysis explained 9% of the overall variance, the second block explained 10% and the third block explained 15%. These findings indicated that a preference for SEOM without condoms and higher levels of sexual arousal when exposed to sexually explicit bareback content were significant predictors of engaging in receptive anal sexual behavior without condoms.

## 4. Discussion

As in other studies researching SEOM use among MSM [2,25,26], SEOM consumption was high, and participants reported spending a substantial amount of time viewing sexually explicit content online. Study participants reported viewing SEOM due to being sexually aroused, wanting to release sexual arousal, being bored, wanting to relieve stress and because of routines/habits [27]. Furthermore, 67% of participants reported consistently viewing SEOM involving bareback content (anal sex without condoms). For most MSM, the primary motives for viewing SEOM without condoms were the fact that bareback sex looked more natural, they depicted sex as if STIs and HIV did not exist, and they made participants feel more sexually aroused.

The study also observed that participants reported significantly higher sexual arousal to picture B (anal sex without a condom) than to picture A (anal sex with a condom). Therefore, presenting sexual arousal while viewing bareback SEOM may positively reinforce partaking in unprotected sex. This could be due to the fact that physiological arousal may be retained along with stimulus content in a learning situation, making MSM operate at a similar cognitive level and, thus, engage in spontaneous processes that may lead to higher risk sexual behaviors [28]. Viewing SEOM depicting unprotected anal sex may more easily activate pleasurable expectations when engaging in sexual activities with a partner. As a result, MSM may experience more sexual/sensorial pleasure associations with condomless sex, causing their sexual behaviors to be more likely to be guided by pleasure-seeking motives, rather than by their knowledge regarding STI risk exposure [29]. This finding was demonstrated by the regression analysis conducted by the present study, showing that preferences for SEOM without condoms and higher levels of arousal when viewing depictions of unprotected anal sex were significant predictors of engaging in receptive anal sex without a condom.

Similar to the observations made by previous studies [2,4,14,23], our participants reported significant exposure to and a preference for bareback SEOM. This could be due to the fact that SEOM may currently contain greater representations of unprotected sex due to the pornography industry’s increased production of bareback SEOM. The increase in condomless SEOM has resulted from a number of factors, including the prevalent dissemination of free access to SEOM on multiple platforms and the increased use of antiretroviral therapy and pre-exposure prophylaxis (PrEP). Antiretroviral therapy has allowed HIV-positive MSM to maintain an undetectable HIV viral load and eliminate any risk of HIV transmission, creating a new “undetectable” identity category with important implications for social and interpersonal interactions [24]. Additionally, the increased use of PrEP has permitted many HIV-negative individuals to significantly reduce their risk of contracting HIV.

It is important to highlight that the study observed differences in engaging in both insertive and receptive unprotected anal sex between gay and bisexual men. The fact that self-identified gay men were more likely to partake in bareback sex in their real-life sexual interactions than self-identified bisexual men, while bisexual men claimed that they more frequently viewed SEOM out of habit, indicated the presence of mediation variables, presumably related to the effects of psychological distress. Several studies have reported that bisexual individuals may be more likely to experience mental health difficulties [30,31] due to the effects of suffering a double stigma from both heterosexuals and gay/lesbians. In addition, bisexuals are also more likely to hide their sexual orientation to protect themselves from discrimination and stigma [32]. This behavior may have negative impacts on their psychological well-being, leading them to view SEOM as a way to mitigate stress and refrain from exposing themselves to unprotected sexual activities out of a fear of being forced to come-out if they were to contract a sexually transmitted infection. Furthermore, the fact that around 10% of participants indicated that they are in a relationship with a woman may also contribute to these differences, making them more likely to use a condom when having extra-relational sex with men to prevent STIs, in addition to contributing to systematic SEOM use to satisfy their homoerotic fantasies.

The literature currently possesses no studies concerning SEOM consumption among Portuguese MSM. This finding reinforces the importance of the present study, as it is possible that cultural variables may also be important in understanding MSM’s responses to SEOM, sexual behavior and its respective influence, since culture transmits values and expectations that influence sexual behaviors. It is critical that we understand the role of cultural influences on behaviors relevant to STI risks, in order to reduce STI prevalence among MSM. Greater insight concerning the influence of SEOM will enrich theoretical risk models that have typically overlooked social and cultural dynamics.

Although Portuguese legislation is generally inclusive of sexual minorities, according to a report from the Council of Europe [33], the identity development of MSM in Portugal is still restricted by negative societal attitudes. These restrictions often result in the internalization of stigma associated with their sexual identity, which has been shown to be associated with increased engagement in risky sexual behaviors [34]. Therefore, our findings demonstrate how personal activities (such viewing SEOM), values and attributed meanings are central to the negotiation of risk in sexual relationships. Sexual risk negotiation can be influenced by a desire for physical pleasure and a dislike of condoms, as well as by role-models depicted in SEOM, who are used to justify unprotected sex among MSM, and may also be influenced by gender stereotypes [35]. The homoerotic-repressive culture and stigma/discrimination that many Portuguese MSM face may pressure them to use SEOM as a way of coping with and obtaining relief from societal repression. In turn, this could lead to a rationalization of sexual risk-taking, while emphasizing fantasies of ‘natural’ (unprotected) sex as a way of producing a sense of ‘normality’, which is usually discouraged by public health messages that portray sexual risks through a biomedical lens.

Perceptions of low levels of sexual control have frequently been used as justifications for engaging in unprotected anal sex in studies among MSM in Portugal. However, previous studies have failed to operationalize SEOM consumption as a possible motive for exposure to higher risk sexual behaviors. Discrimination against homosexual behaviors creates a context of risk and oppression, which have been found to be strong predictors of HIV transmission risk among MSM. However, this study’s findings encourage further consideration of the fact that men might seek out what they view as more pleasurable and ‘natural’ sexual activities, both online and in their sexual interactions, despite the presence of associated risks, including those related to STI and HIV infection. Therefore, SEOM consumption needs to be taken into consideration when developing sexually transmitted infection prevention strategies.

### Study Limitations

Several limitations ultimately restrict the ability to generalize the research findings. The study sample was disproportionately comprised of urban, well-educated men, who possessed Internet and technological access, and who were recruited through social organizations and social networks. Consequently, the extent to which these MSM are representative of MSM from other regions of Portugal is uncertain. However, the intention of this study was not to generalize its findings to all Portuguese MSM, but, rather, to contribute to better understanding the associations between SEOM use and sexual behaviors among MSM in Portugal. Moreover, it is important to highlight that this study relied on self-reported and subjective assessments that are both subject to social desirability bias. Social desirability bias was likely minimized by the anonymous nature of our survey, although it is still possible that there may have been some underreporting of behaviors considered to be more socially undesirable. The relationship between SEOM use and sexual behaviors has been the subject of greater research in the United States, thus, further comparative research across European countries and beyond could enrich the international literature regarding this topic. Furthermore, this study recommends that future studies utilize longitudinal and experimental study designs, in order to assess any causal relationships among study variables.

## 5. Conclusions

Overall, our findings suggest that Portuguese MSM frequently view SEOM and that they possess a stated preference for SEOM depicting unprotected anal sex. Additionally, the research revealed that self-identified gay men more frequently engaged in unprotected anal sex than self-identified bisexual men. Moreover, the research findings indicated that a preference for viewing and higher levels of arousal when viewing SEOM showing unprotected sex are significant predictors of partaking in receptive anal sex without condoms. Therefore, the positive association between bareback SEOM consumption and engaging in unprotected anal sex should provide an opportunity for MSM-targeted STI/HIV online prevention strategies tailored to Portuguese MSM. Specifically, many websites and apps now offer men SEOM containing bareback content, along with links to find sexual partners online. The association between SEOM consumption and sexual risk-taking is a reality, and, as a result, this relationship should be studied more thoroughly, taking into account the possible influence of hostile social contexts on these behaviors. Public health efforts seeking to decrease the negative influences of SEOM could have the potential to increase sexual quality of life, while decreasing sexual risk-taking among MSM in Portugal.

## Figures and Tables

**Table 1 behavsci-11-00038-t001:** Sociodemographic Characteristics of the Sample Participants (N = 1577).

Variables	Categories	N	%	M	SD
Age				35.69	11.16
Frequency of sexual intercourse (per week)				3.21	3.10
Sexual orientation	Gay	1061	67.3		
	Bisexual	516	32.7		
Place of residence	Large urban environment	803	50.9		
	Small urban environment	506	32.1		
	Large rural environment	99	6.3		
	Small rural environment	169	10.7		
Marital status	Single	899	57.0		
	Married to a man	21	1.3		
	Married to a woman	129	8.2		
	De facto union with a man	129	8.2		
	De facto union with a woman	39	2.5		
	Dating a man	231	14.6		
	Dating a woman	50	3.2		
	Divorced from a man	9	0.6		
	Divorced from a woman	70	4.4		
Educational attainment	Up to 12 years of schooling	479	30.4		
	University degree	1098	69.6		
Occupation	Student	230	14.6		
	Unemployed	191	12.1		
	Employed	793	50.3		
	Self-employed	181	11.5		
	Retired	61	3.8		
	Temporary	40	2.5		
	Other	81	5.1		

**Table 2 behavsci-11-00038-t002:** Sexual Practices by Sexual Orientation.

Sexual Activity	Sexual Orientation	M	SD	t(df)	*p*
Receptive anal sex with condoms	Gay	2.68	1.58	−0.056(147)	0.956
	Bisexual	2.70	1.68		
Receptive anal sex without condoms	Gay	2.23	1.47	3.275(148)	0.001 *
	Bisexual	1.45	0.93		
Insertive anal sex with condoms	Gay	2.55	1.74	−1.656(145)	0.100
	Bisexual	3.06	1.70		
Insertive anal sex without condoms	Gay	2.04	1.43	1.957(146)	0.049 *
	Bisexual	1.58	1.06		

* <0.05.

**Table 3 behavsci-11-00038-t003:** SEOM Use by Sexual Orientation.

Variables	Categories	Sexual Orientation	*M*	*SD*	*t(df)*	*p*
Average length of SEOM viewing sessions		Gay	73.40	148.16	0.222(144)	0.824
	(in min)	Bisexual	68.36	58.26		
Motives for SEOM use (1 = Not true; 3 = Totally true)	Because I am very sexually aroused	Gay	2.42	0.66	0.173(151)	0.863
		Bisexual	2.40	0.68		
	Because I want to release sexual arousal	Gay	2.35	0.75	−0.441(151)	0.660
		Bisexual	2.41	0.71		
	To entertain myself because I am bored	Gay	2.00	0.78	−1.515(148)	0.132
		Bisexual	2.21	0.75		
	To relieve stress	Gay	2.03	0.77	−1.162(147)	0.247
		Bisexual	2.19	0.71		
	Because it is a habit or a routine	Gay	1.75	0.76	−2.115(145)	0.036 *
		Bisexual	2.04	0.78		
	Just curious, in a non-sexual way	Gay	1.65	0.75	−1.648(147)	0.102
		Bisexual	1.86	0.74		
	To help me fall asleep faster	Gay	1.72	0.76	0.019(144)	0.985
		Bisexual	1.71	0.77		
	To help create a sexy environment for me and my partner	Gay	1.49	0.73	0.884(143)	0.378
		Bisexual	1.37	0.64		
	To learn new sexual techniques or positions	Gay	1.65	0.78	−0.770(145)	0.443
		Bisexual	1.76	0.79		
Percentage of Condomless SEOM use (1–10)		Gay	6.71	20.82	0.275(147)	0.783
		Bisexual	6.57	20.95		
Motives for viewing bareback SEOM (without condoms) (1 = Not true; 3 = Totally true)	Looks more natural	Gay	2.14	0.83	0.094(144)	0.926
		Bisexual	2.12	0.82		
	It is similar to what I do in my sex life	Gay	1.75	0.84	1.708(142)	0.040 *
		Bisexual	1.51	0.69		
	Allows me to fantasize about a behavior I never engage in	Gay	1.77	0.80	−1.110(143)	0.269
		Bisexual	1.93	0.77		
	Reminds me of what sex could be like if there were no STIs or HIV	Gay	2.17	0.82	−0.302(142)	0.763
		Bisexual	2.21	0.78		
	Turns me on more	Gay	2.21	0.81	0.387(139)	0.699
		Bisexual	2.15	0.89		
	I like feeling the risk the actors may be taking	Gay	1.49	0.72	−0.041(141)	0.967
		Bisexual	1.50	0.65		
	I like watching anal sex with or without a condom	Gay	2.32	0.76	2.195(140)	0.030 *
		Bisexual	2.02	0.77		
Level of arousal reacting to direct stimuli (pictures of anal sex) (1–10)	Image A (anal sex with a condom)	Gay	6.42	2.14	−0.902(152)	0.369
		Bisexual	6.76	2.29		
	Image B (anal sex without a condom)	Gay	7.85	2.62	1.484(152)	0.140
		Bisexual	7.14	2.97		

* <0.05.

**Table 4 behavsci-11-00038-t004:** SEOM Consumption by Sexual Orientation.

Variables	Categories	Sexual Orientation		Chi(df)	*p*
		Gay	Bisexual		
Frequency of SEOM use	Every day	22.7%	13.3%	4.572(3)	0.206
	5–6 days per week	11.3%	7.3%		
	3–4 days per week	14.0%	4.7%		
	1–2 days per week	21.3%	5.3%		
Post-SEOM use activity	Nothing	7.3%	5.3%	1.580(2)	0.454
	Masturbation	59.6%	24.5%		
	Sex with a partner	2.6%	0.7%		
Type of SEOM viewed	Videos	60.5%	22.4%	8.704(3)	0.034 *
	Photographs	2.6%	5.3%		
	Webcam/live stream	5.9%	2.0%		
	Erotic stories	0.7%	0.7%		
SEOM condom use preferences	Without condoms	27.6%	9.9%	0.923(2)	0.630
	With condoms	9.9%	5.3%		
	No preference	31.6%	15.8%		

* <0.05.

**Table 5 behavsci-11-00038-t005:** Correlation Matrix between Images A and B and Anal Sexual Behaviors.

	1	2	3	4	5	6
1—Image A (anal sex with a condom)	1					
2—Image B (anal sex without a condom)	0.160 *	1				
3—Receptive anal sex with condoms	0.052	−0.187 *	1			
4—Receptive anal sex without condoms	−0.102	0.251 **	−0.091	1		
5—Insertive anal sex with condoms	0.143	−0.117	0.369 **	−0.269 **	1	
6—Insertive anal sex without condoms	−0.005	0.192 *	−0.194 *	0.501 **	−0.095	1

* <0.05; ** <0.001.

**Table 6 behavsci-11-00038-t006:** Hierarchical Multiple Regression Analysis Predicting Receptive Anal Sex Without Condoms.

Variables	Model 1			Model 2			Model 3		
	*B*	*SEB*	*ß*	*B*	*SEB*	*ß*	*B*	*SEB*	*ß*
Age	0.002	0.011	0.019	0.003	0.011	0.026	0.001	0.011	0.005
Preference for Bareback SEOM	−0.463	0.134	−0.304 *	−0.491	0.135	−0.322 *	−0.384	0.140	−0.252 *
Time spent using SEOM				−0.001	0.001	−0.105	−0.001	0.001	−0.099
Arousal to Image A							−0.089	0.065	−0.122
Arousal to Image B							0.107	0.050	0.194 *
*R* ^2^	0.092			0.103			0.148		
*F for change in R* ^2^	11.189 *			8.317 *			7.178 *		

* <0.05.

## Data Availability

The data presented in this study are available upon request.

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
