# Peer review of "Sexually Explicit Online Media Use and Sexual Behavior among Sexual Minority Men in Portugal"

_behavsci, 2021, doi:10.3390/bs11030038_

Round 1

Reviewer 1 Report

This reviewer commends the authors for their study: 

  • The background should include epidemiology of  high risk-sexual behavior, including adult consumption of SEOM, and the broad scope of factors that influence it.
  • The inclusion of the prevalence of consumption of SEOM by the adult population in general and the study population in particular would be important.
  • What is the research question? Please articulate clearly.
  • What is the variance explained by each of the predictor’s of receptive anal sex without condoms ?
  • Some of the  study findings could be depicted using illustrations other than tables.
  • What would be the authors' public health policy or programmatic intervention recommendations based on their study? Prevention of consumption of SEOM? Prevention of high risk-sexual behavior?

Reviewer 2 Report

I really appreciated the article “Sexually Explicit Online Media Use and Sexual Behavior in Sexual Minority Men in Portugal.” It is well written and interesting. Nonetheless, I think that the paper would benefit from some minor modifications.

Introduction

I really appreciated the clarity of the introduction. It is short and concise but effective.

Methods

  • It is my understanding that the questions about the SEOM are all open-ended. If so, please specify. Some of the questions may also be formulated as close-ended questions. For example “On average, how much time do you spend viewing SEOM?” was formulated as an open question as well?
  • I suggest the authors divide the method section into paragraphs: participants, procedure, and measures, for the sake of clarity.

Results

  • I wonder if there are differences between younger and older participants in terms of frequency of sex, SEOM use. I don’t know if the authors thought about or did a comparison based on age.
  • I would move the inclusion criteria in the method section.
  • I don’t really understand how the content of the open-ended (?) questions was carried out. Throughout a content analysis? a thematic analysis? Who did this analysis? It was calculated inter-rater reliability? Or did the authors used close-ended questions? In that case, how did they choose the answers?

Conclusion

  • I would suggest adding a limitations section.

Reviewer 3 Report

This article suggests a current and attractive topic for the academy. The research is timely and worthwhile. The research problem is clearly defined. The authors provide fresh insight into the field.

I hope you find the following observations helpful:

Structure: Articles should be reformatted according to a standard structure, which is set out in the instructions for authors of the journal (sections are Introduction, Materials and Methods, Results, and Discussions, Conclusion). See new template. The abstract should be improved.

Results: Perhaps it is better to visualize in more charts based on statistical methods of calculation. In my opinion, it may be better to provide the results of testing these methods (if any) in the Results section.

The authors should be appropriate to explain the choice of methodology a little better. It would be advisable to extend the literature review.

Need to revise and check citations in the text and in the references section. I suggest you add this reference: Zakharchenko O., Zakharchenko A., Fedushko S. Global Challenges Are Not For Women: Gender Peculiarities Of Content In Ukrainian Facebook Community During High-Involving Social Discussions. CEUR Workshop Proceedings. Vol 2616: Proceedings of the 2nd International Workshop on Control, Optimisation and Analytical Processing of Social Networks (COAPSN-2020), Lviv, Ukraine, May 21, 2020. p. 101-111. http://ceur-ws.org/Vol-2616/paper9.pdf

Overall, I find the paper adequate but it can be improved by addressing the aforementioned issues. Especially the problem of the paper structure.

Congratulations on a job well done.
